# Predictors of early initiation of breastfeeding in Indonesia: A population-based cross-sectional survey

Maria Gayatri[1]*, Gouranga Lal Dasvarma[2]

1 National Population and Family Planning Board—Family Planning and Family Welfare Research and Development Unit, East Jakarta, Indonesia, 2 Flinders University—College of Humanities, Arts and Social Sciences, Adelaide, South Australia, Australia

* maria.gayatri.bkkbn@gmail.com

## Abstract

### Introduction

Commencing breastfeeding within one hour of birth is defined as early initiation of breast-feeding (EIBF). Both the mother and child benefit from EIBF. This study aims to identify the predictors of EIBF among Indonesian women.

### Methods

This paper analyses data from a weighted sample of 6,616 women collected at the Indonesia Demographic and Health Survey (IDHS) 2017.The frequency of EIBF is measured by the proportion of children born in the two years preceding the survey who received breast-milk within one hour of birth. The analysis uses bivariate and multivariate logistic regression for complex sample designs, adjusted for confounders to examine the relationship of EIBF with women's individual, household and community level characteristics.

### Results

Overall, 57% (95% CI: 54.9%-58.2%) of the children born in the two years preceding the survey had EIBF. Statistically significant ($p<0.05$) predictors of EIBF are women's non-working status, second or higher order of the birth of the most recent child, average or large size of the most recent child at birth, poor status of the household and non-agricultural work of the woman's husband; while statistically highly significant ($p<0.01$) predictors are skin-to-skin contact with the new-born (OR: 2.62; 95% CI: 2.28–3.00), Caesarean deliveries (OR: 0.47; 95% CI: 0.40–0.56), and skilled birth attendants (OR: 1.83; 95% CI: 1.65–2.08). Caesarean deliveries reduce the likelihood of EIBF by half compared to vaginal deliveries. Women's age, education or rural-urban residence display no statistically significant relationship with EIBF.

**Data Availability Statement:** All data files are available from the demographic and health survey database https://dhsprogram.com.

**Funding:** This study was supported by Indonesian National Population and Family Planning Board

(Family Planning and Family Welfare Research and Development Unit) in the form of funds in Accredited Scientific Journal Publications to MG (Code: 521219) and Flinders University (College of Humanities, Arts and Social Sciences) in the form of funds in the Consultancy Account of GLD (Cost Centre/Project: 422.30877). The funders had no role in study design, data collection and analysis, decision to publish, or preparation of the manuscript.

**Competing interests:** No authors have competing interests.

## Conclusion

Skin-to-skin contact, mode of delivery and type of birth attendance exert the strongest influence on EIBF in Indonesia in 2017. EIBF should be continuously promoted and supported particularly among mothers who do not have early skin-to-skin contact with their new-born, who have Caesarean deliveries and who have no skilled birth attendant.

## Introduction

Breastfeeding is a cost-effective investment for reducing infant and child morbidity and mortality [1]. Breastmilk is the first natural food for infants during the first six months of life which meet their nutritional requirements for optimal growth, development and health [2]. In 1989 the World Health Organisation and United Nations Children's Fund (WHO/UNICEF) presented to the world a joint statement on the protection, promotion and support of breastfeeding through an initiative known as *Ten Steps to Successful Breastfeeding* [3]. This was ratified at the Innocenti Declaration in 1990 which called upon the world to fully implement the Ten Steps in all maternities by 1995 with the aim of improving breastfeeding practices and helping mothers to initiate breastfeeding within one hour of birth.

WHO and UNICEF recommend that children should be given mother's breast milk within the first hour of birth so that they can receive colostrum, the first milk which is extremely rich in nutrients and antibodies [4,5]. The practice of initiating breastfeeding within the first hour of birth, known as Early Initiation of Breastfeeding (EIBF) benefits both the mothers and their children. It reduces mortality among infants through increased exclusive breastfeeding and associated mechanisms [6] by allowing the infants to get the highly nutritious maternal colostrum that reduces the risk of microbial translocation, accelerates intestinal maturation and promotes resistance to infection [6–10]. EIBF also reduces the risk of hypothermia and helps develop bonding between mother and baby through close physical contact [9]. On the other hand, EIBF helps the mother by stimulating breast milk production and facilitating the release of oxytocin which is important for the contraction of the uterus and reduction of postpartum haemorrhage [11]. Women who delay the initiation of breastfeeding (delayed initiation of breastfeeding (DIBF) is defined as initiating breastfeeding later than one hour of birth) face an increased risk of their child dying in the neonatal period [5,9]. Thus, EIBF is crucial for the survival of both the mother and the child.

Global estimates for 2017 show that only 42% of all new born babies are put to the mother's breast within one hour of delivery [4]. The rates of EIBF vary across the world from 35% in the Middle East and North Africa to about 65% in Eastern and Southern Africa [4]. In Indonesia, it is estimated that 57% of new-born children were breastfed within the first hour of birth in 2017 [12]. This leaves a massive 43% of new-born children who were not breastfed within the first hour of birth. Therefore, it is very important to identify the predictors of EIBF so that this large gap between the prevalence of early and delayed initiation of breastfeeding can be reduced and programs for designing and promoting early breastfeeding can be developed for the nation. Several studies have identified a variety of factors that may influence the practice of EIBF. Among the demographic factors, women's age [13] and birth order of the child [14–16] are documented as significant predictors of EIBF. Health related factors for EIBF include access to antenatal care [4], vaginal delivery of babies [4,11,16–22], delivery of babies assisted by skilled birth attendants [4,16,17,23], the practice of skin-to-skin contact between mother and baby soon after birth [16,21] and the baby's size at birth (proxy for birthweight) [16,18].

Other factors such as distribution of breastmilk substitutes [2,24] have also been found to be related to the adoption or non-adoption of EIBF.

The present study aims to identify the predictors of EIBF among Indonesian women through an analysis of data collected at the Indonesia Demographic and Health Survey (IDHS) 2017.

## Materials and methods

This study uses data collected at IDHS 2017, a nationally representative cross-sectional survey jointly conducted by the National Population and Family Planning Board, Statistics Indonesia and the Ministry of Health with technical assistance from the Demographic and Health Surveys (DHS) Program [12]. IDHS 2017 used a two-stage stratified sampling design. The first stage consisted of selecting 1,970 census blocks from urban and rural areas of Indonesia by systematic sampling proportional to size, where the size of each census block comprises the number of households listed at the 2010 Indonesian Population Census. The second stage involved the selection of 25 households by systematic sampling from each selected census block [12].

The present study utilises data collected by direct interviews with structured questionnaires (women's questionnaires) administered to a weighted sample of 6,616 women at the IDHS 2017 and analyses information with respect to their most recent child born in the last two years preceding the survey. The interviews were conducted during July-September 2017.

### Measurements

**Dependent variable.**   The dependent variable in this study is EIBF, which is based on the women's report, in terms of number of hours or days after delivery when they started breastfeeding their babies. The dependent variable is categorised in a binary form, namely EIBF or DIBF. EIBF is measured as the proportion of infants aged 0–23 months who were put to their mother's breast within one hour of birth [25].

**Predictor variables.**   The predictor or independent variables in this study comprise the characteristics of the sample of women interviewed at IDHS 2017. These variables are classified as individual, household and community level variables.

Individual level variables include the women's socio-demographic characteristics such as age at delivery classified in groups (15–19, 20–29, 30–39 or 40–49 years), education (no education, primary, and secondary or higher education), and work status (working or not working).

The variables at the household level consist of socio-economic and health related characteristics such as birth order of the most recent child (1, 2, 3, or 4 or more), sex of the most recent child (male or female), education of child's father (no education, primary, secondary or above), occupation of the child's father (not working, working in agriculture or working in non-agriculture) and household wealth index (poor, middle or rich). Household wealth index is computed as a composite index using Principal Components Analysis of variables about a household's ownership of assets and facilities such as access to electricity, ownership of household durable goods such as television, radio, telephone, refrigerator, motorcycle and private car, materials used for housing construction such as flooring materials, rooms used for sleeping, place for cooking and cooking fuel, source and treatment of drinking water, and sanitation facilities [26,27]. Based on the guide to DHS statistics by the DHS Program, each household asset for which information is available is assigned a weight or factor score generated by the Principal Components Analysis and the scores are then transformed into standard normal variates with a mean of zero and standard deviation of one [26,27]. The standardised scores are used to create cut-offs defining five wealth quintiles called Lowest or Poorest, Second or Poorer, Middle, Fourth or Richer and Highest or Richest. However, for purposes of the present

study these five categories of household wealth index have been recoded into three groups, namely Poor, Middle and Rich by combining the first two as Poor and the last two as Rich. Individuals residing in a household are assigned the overall score of the household.

Health related factors at the household level such as skin-to-skin contact between the women and their new-born babies after birth (yes or no), antenatal care (ANC) visits by the women (no ANC visit, 1–3 ANC visits or 4 or more ANC visits), information on the child's size at birth (small, average or large), mode of delivery (normal vaginal delivery or Caesarean or C-section), place of delivery (respondent's home or health facility—such as hospital, village maternity post, clinic, primary health centre, village midwife and private medical sector) and type of assistance during delivery (unskilled birth attendant-such as traditional birth attendant, relative or friends, and skilled birth attendant/health personnel—such as general practitioner, obstetrician, nurse, midwife and village midwife).

The community level variable consists of the woman's place of residence (urban or rural). A list of all the variables, with their respective categories is given in Table 1.

## Analysis

The data have been analysed in two steps. The first step consists of descriptive analysis showing the distribution of the sample of women according to the variables described above and the women's EIBF status. The proportion of women who initiated early breastfeeding is reported in percentages. The first inferential statistics used in this study are based on bivariate relationships between EIBF and each variable by using bivariate logistic regression. The statistically significant variables in the bivariate logistic regression are included in the second step where the determinants of EIBF are analysed by using a multivariate logistic regression in order to examine the relationship between EIBF and various predictor variables.

The particular multivariate logistic regression used in the second step is complex samples logistic regression, which is the appropriate method for a binary dependent variable like EIBF calculated from samples drawn by complex sampling methods [28], such as that used in IDHS 2017. In this method, the sample must be weighted to inflate the sample data, either from a small province or a large province to be equivalent to the size of the total national population. Weights are used in the survey in order to consider the sampling probabilities, and adjust for nonresponse error or coverage error [29]. After weighting the data, the proportion of sampled women in each province becomes equivalent to the actual number of women in that province [12]. A complex sample provides subpopulation analysis of survey data where the study population integrates weights, stratification and clustering design [30,31]. IDHS 2017 is a geographically stratified and clustered sample of households with the number of women in each province representing the population of that province [12]. A complex sample produces accurate point estimates and standard errors associated with model parameters and odds ratios because it accounts for the complex design of the sample by considering Primary Sampling Units (similar to cluster number and/or the ultimate area unit), stratification (similar to the basic geographic units) and sampling weight [27,29,32]. The logistic regression model for survey data estimates the parameter β by using pseudo-maximum likelihood [32], where β measures the odds of the dependent variable being true for a particular predictor variable relative to the reference category of that variable. Logistic regression using a complex sampling design produces a good power of the model to show the strength of the association between the covariates and outcome variables from a Bernoulli distribution as shown in formula (1) [32].

$$y \sim \text{Bernoulli} \left( P(y = 1) = \frac{\exp\{\beta_0 + \beta_1 x_1 + \beta_2 x_2 + \cdots + \beta_n x_n\}}{1 + \exp\{\beta_0 + \beta_1 x_1 + \beta_2 x_2 + \cdots + \beta_n x_n\}} \right) \tag{1}$$

**Table 1. Characteristics of women having their most recent birth in the two years preceding the survey according to initiation of breastfeeding status, Indonesia 2017.**

| Characteristics | Women giving birth to their most recently born children during the 2 years preceding the survey | | | |
| --- | --- | --- | --- | --- |
| | Early initiation of breastfeeding (n = 3,742) | | Delayed initiation of breastfeeding (n = 2,874) | |
| | Number | % | Number | % |
| Mother's age at childbirth | | | | |
| • 15–19 | 149 | 51.3 | 143 | 48.7 |
| • 20–29 | 1,707 | 56.0 | 1,343 | 44.0 |
| • 30–39 | 1,651 | 58.0 | 1,197 | 42.0 |
| • 40–49 | 235 | 44.8 | 191 | 55.2 |
| Mother's education | | | | |
| • No education | 40 | 61.5 | 25 | 38.5 |
| • Primary | 872 | 57.0 | 659 | 43.0 |
| • Secondary or higher | 2,831 | 56.4 | 2,189 | 43.6 |
| Mothers' working status | | | | |
| • Not working | 2,136 | 57.8 | 1,557 | 42.2 |
| • Working | 1,606 | 54.9 | 1,317 | 45.1 |
| Birth order of the child | | | | |
| • First | 1,094 | 50.5 | 1,074 | 49.5 |
| • Second or third | 2,158 | 60.2 | 1,426 | 39.8 |
| • Fourth or more | 490 | 56.8 | 374 | 43.2 |
| Sex of the child | | | | |
| • Male | 1,921 | 56.1 | 1,505 | 43.9 |
| • Female | 1,821 | 57.1 | 1,369 | 42.9 |
| Father's education | | | | |
| • No education | 35 | 56.5 | 27 | 43.5 |
| • Primary | 948 | 56.9 | 718 | 43.1 |
| • Secondary and above | 2,760 | 56.5 | 2,128 | 43.5 |
| Fathers' occupation | | | | |
| • Not working | 13 | 34.2 | 25 | 65.8 |
| • Agriculture | 759 | 54.1 | 645 | 45.9 |
| • Non-agriculture | 2,970 | 57.4 | 2,204 | 42.6 |
| Wealth index of family | | | | |
| • Poor | 1,547 | 58.1 | 1,115 | 41.9 |
| • Middle | 747 | 57.2 | 559 | 42.8 |
| • Rich | 1,448 | 54.7 | 1,200 | 45.3 |
| Skin-to-skin contact | | | | |
| • No | 1,114 | 41.2 | 1,588 | 58.8 |
| • Yes | 2,628 | 67.2 | 1,286 | 32.8 |
| Mother's ANC visits | | | | |
| • No ANC visit | 90 | 51.1 | 86 | 48.9 |
| • 1–3 ANC visits | 245 | 51.8 | 227 | 48.2 |
| • 4 or more ANC visits | 3,407 | 57.1 | 2,561 | 42.9 |
| Child's size at birth | | | | |
| • Small | 364 | 47.3 | 406 | 52.7 |
| • Average | 2,227 | 58.2 | 1,598 | 41.8 |
| • Large | 1,151 | 57.0 | 870 | 43.0 |
| Mode of delivery | | | | |
| • Normal vaginal delivery | 3,274 | 61.2 | 2,074 | 38.8 |

(*Continued*)

**Table 1.** (Continued)

| Characteristics | Women giving birth to their most recently born children during the 2 years preceding the survey | | | |
|---|---|---|---|---|
| | Early initiation of breastfeeding (n = 3,742) | | Delayed initiation of breastfeeding (n = 2,874) | |
| | Number | % | Number | % |
| • C-section | 467 | 36.8 | 801 | 63.2 |
| Place of delivery | | | | |
| • Home | 618 | 56.6 | 473 | 43.4 |
| • Health Facility | 3,125 | 56.6 | 2,400 | 43.4 |
| Assistance during delivery | | | | |
| • Unskilled birth attendant | 264 | 54.9 | 216 | 45.1 |
| • Skilled birth attendant | 3,478 | 56.7 | 2,658 | 43.3 |
| Area of residence | | | | |
| • Urban | 1,829 | 56.8 | 1,390 | 43.2 |
| • Rural | 1,914 | 56.3 | 1,483 | 43.7 |

Source: Computed by the authors from IDHS 2017 data

The final model includes explanatory variables for which the level of significance is less than or equal to 5% (p ≤ 0.05). Odds ratios (OR) are reported with 95% confidence interval (95% CI). Sampling weights are applied to all the analyses. The statistical analysis is performed by using the computer software Stata Version 15 (Stata Corp, College Station, TX), and all the analyses are weighted and assigned values using the '*svy*' survey prefix command for complex samples [30,31].

## Ethics approval

The data have been downloaded by registering with the Demographic and Health Surveys (DHS) website (https://dhsprogram.com). According to the DHS Program, "the procedures and questionnaires for standard DHS surveys are reviewed and approved by The Institutional Review Board (IRB) of ICF International while country-specific DHS protocols are reviewed by the IRB of ICF International and typically by an IRB in the host country" [33]. The IRB of ICF International ensures that the survey complies with the U.S. Department of Health and Human Services regulations for the protection of human subjects, while the host country IRB ensures that the survey complies with laws and norms of the nation. Moreover, the names and addresses of the respondents are not included while downloading the data. Therefore, no separate ethics approval is required for using the IDHS 2017 data for this paper.

## Results

### Descriptive analysis

Table 1 displays the characteristics of the 6,616 women aged 15–49 years who gave birth in the two years preceding IDHS2017, according to their EIBF status. Of these 6,616 women 3,742 (56.6%) reported practising EIBF (95% CI: 54.9%-58.2%).

Younger women reported a higher prevalence of EIBF than older women. For example, 51%, 56% and 58% of the women aged 15–19, 20–29 and 30–39 years respectively initiated breastfeeding early compared to 45% of women aged 40–49 years. Women's education is found to be inversely associated with EIBF—the higher the level of a woman's education, the lower is the percentage practising EIBF. However, it is interesting to find that the education of the child's father does not make a difference in EIBF, with all the three categories of father's

education showing an EIBF of 57%. Women's work status is found to make some difference in EIBF with non-working women exhibiting a slightly higher prevalence of EIBF (58%), compared to working women (55%). However, the opposite appears to be true with respect to the work status of the child's father, with a lower prevalence of EIBF (34%) among non-working fathers, compared to a higher prevalence of EIBF among fathers working in agriculture (54%) or non-agricultural activities (57%). The second or third birth order children tend to have more chances of early breastfeeding with an EIBF of 60%, compared to the first-born children (EIBF 51%) or the fourth and later born children (EIBF 57%). Sex of the child does not appear to matter in terms of EIBF or DIBF as almost equal proportions (56% and 57% respectively) of the male and female children are breastfed early. The number of antenatal care (ANC) visits appears to influence EIBF only very moderately. Ninety one percent of the women (3,407 of 3,742, Table 1) reported making four or more ANC visits, but they have only a slightly higher prevalence of EIBF (57%) compared to women with no ANC visit (51%) or women with 1–3 ANC visits (53%). Women with small sized babies at birth tend to practise EIBF less than women with average or large sized babies at birth, the respective percentages of EIBF being 47%, 58% and 57%. An average size of the baby at birth, indicating average birthweight appears to have the best chance of being initiated into early breastfeeding. Women who are assisted by skilled birth attendants during delivery are only a little better in practising EIBF (57%) compared women who are not assisted by skilled birth attendants (EIBF 55%). Women having skin-to-skin contact with their babies are found to have a higher prevalence of EIBF (67%) than women who do not have skin-to-skin contact (41%). Place of delivery (home or a health facility) and women's place of residence (rural-urban) are found to have no influence on EIBF, as both categories of each of these two variables show an EIBF prevalence of around 56% and 57%.

## Multivariate analysis

Table 2 shows the results of multivariate logistic regression analysis. Only the statistically significant relationships showing adjusted odds ratios (Adj OR) with $p \leq 0.05$ are described in this section. Children of working mothers are 11% less likely to have EIBF compared to children of non-working mothers, showing an inverse association between EIBF and mothers' working status. With respect to the child's birth order, the analysis shows that the children of the second and third, and fourth or higher birth order are respectively 45% and to 34% more likely to have an EIBF compared to the first-born children. The working status of the child's father exhibits an association with EIBF that is opposite to the association between mother's working status and EIBF. Thus, the children of working fathers have a greater chance of EIBF compared to children of non-working fathers, but only the father's non-agricultural work exhibits a statistically significant association with EIBF (Adj OR: 2.29; 95% CI: 1.16–4.53). The relatively large CI of this adjusted OR is worth noting. Household wealth index is inversely related to EIBF, but a statistically significant relationship between the two is found only between rich wealth index and EIBF, where women living in rich households are 15% less likely to initiate early breastfeeding compared to women living in poor households.

Children who have skin to skin contact with their mothers soon after birth are about two and a half times more likely to have an EIBF compared to children who do not have skin to skin contact. The child's size at birth is positively related to EIBF, as it is found that children with average or large sizes at birth are 40% more likely to have an EIBF compared to children with small sizes at birth. The mode of delivery has a considerable influence on EIBF as it found that women delivering with a C-section are 53% less likely to initiate early breastfeeding compared to women delivering through vaginal birth. Women having skilled assistants at birth

**Table 2. Unadjusted and adjusted odds ratios of predictors of early initiation of breastfeeding.**

| Covariates | Breastfeeding initiation within 1 hour of birth | | | |
| --- | --- | --- | --- | --- |
| | Unadjusted OR (95% CI) | p-value | Adjusted OR (95% CI) | p-value |
| Mother's age at childbirth | | | | |
| • 15–19 | 1 | | | |
| • 20–29 | 1.21 (0.90, 1.62) | 0.206 | - | - |
| • 30–39 | 1.31 (0.98, 1.75) | 0.068 | - | - |
| • 40–49 | 1.17 (0.82, 1.68) | 0.388 | - | - |
| Mother's education | | | | |
| • No education | 1 | | | |
| • Primary | 0.80 (0.46, 1.40) | 0.439 | - | - |
| • Secondary or higher | 0.79 (0.45, 1.37) | 0.393 | - | - |
| Maternal working status | | | | |
| • Not working | 1 | | | |
| • Working | 0.89 (0.79, 1.00) | 0.051 | 0.87 (0.71, 0.96) | 0.022 |
| Birth order of child | | | | |
| • First | 1 | | | |
| • Second or third | 1.48 (1.30, 1.69) | 0.000 | 1.45 (1.27, 1.67) | 0.000 |
| • Fourth or more | 1.29 (1.06, 1.56) | 0.009 | 1.34 (1.09, 1.65) | 0.006 |
| Sex of child | | | | |
| • Male | 1 | | | |
| • Female | 1.04 (0.93, 1.17) | 0.490 | - | - |
| Father's education | | | | |
| • No education | 1 | | | |
| • Primary | 1.02 (0.58, 1.79) | 0.948 | - | - |
| • Secondary and above | 1.00 (0.57, 1.76) | 0.994 | - | - |
| Father's occupation | | | | |
| • Not working | 1 | | | |
| • Agriculture | 2.30 (1.09, 4.84) | 0.028 | 1.85 (0.94, 3.65) | 0.074 |
| • Non-agriculture | 2.63 (1.25, 5.54) | 0.011 | 2.29 (1.16, 4.53) | 0.017 |
| Wealth index | | | | |
| • Poor | 1 | | | |
| • Middle | 0.96 (0.81, 1.14) | 0.664 | 0.91 (0.76, 1.09) | 0.300 |
| • Rich | 0.87 (0.75, 1.00) | 0.050 | 0.86 (0.73, 0.98) | 0.044 |
| Skin-to-skin contact | | | | |
| • No | 1 | | | |
| • Yes | 2.91 (2.57, 3.31) | 0.000 | 2.62 (2.28, 3.00) | 0.000 |
| Mother's ANC visits | | | | |
| • No ANC visit | 1 | | | |
| • 1–3 ANC visits | 1.03 (0.68, 1.57) | 0.680 | - | - |
| • 4 or more ANC visits | 1.27 (0.89, 1.83) | 0.890 | - | - |
| Child size at birth | | | | |
| • Small | 1 | | | |
| • Average | 1.55 (1.30, 1.85) | 0.000 | 1.44 (1.19, 1.74) | 0.000 |
| • Large | 1.47 (1.21, 1.80) | 0.000 | 1.39 (1.13, 1.72) | 0.002 |
| Mode of delivery | | | | |
| • Normal vaginal delivery | 1 | | | |
| • C-section | 0.37 (0.31, 0.43) | 0.000 | 0.47 (0.40, 0.56) | 0.000 |
| Place of delivery | | | | |

*(Continued)*

**Table 2.** (Continued)

| Covariates | Breastfeeding initiation within 1 hour of birth | | | |
|---|---|---|---|---|
| | Unadjusted OR (95% CI) | p-value | Adjusted OR (95% CI) | p-value |
| • Home | 1 | | | |
| • Health Facility | 1.00 (0.86, 1.17) | 0.995 | - | - |
| Assistance during delivery | | | | |
| • Unskilled birth attendant | 1 | | | |
| • Skilled birth attendant | 1.77 (1.30, 2.24) | 0.046 | 1.83 (1.65, 2.08) | 0.016 |
| Area of residence | | | | |
| • Urban | 1 | | | |
| • Rural | 0.98 (0.86, 1.12) | 0.778 | - | - |

Source: Calculated by the authors from IDHS 2017 data

have a much better chance of practising EIBF than women with unskilled birth attendants. This is shown by the fact that children born with skilled birth attendants have their chances of EIBF increase by more than 80% compared to children born without any skilled birth attendants.

Women's age at childbirth, sex of the child, women's education, education of father of the children, the number of ANC visits, place of delivery and area of residence (rural-urban) do not show any statistically significant relationship with EIBF and, as mentioned earlier their adjusted odds ratios are not shown in Table 2.

## Discussion

Globally, only 45% of newborn babies are put to their mother's breast within the first hour of birth [10]. The present study shows that nearly 57% of the last-born infants in Indonesia are breastfed within the first hour of birth. Over the last three Indonesia Demographic and Health Surveys (IDHS's), spanning a period of 10 years from 2007 to 2017, the prevalence of children breastfed within the first hour of birth has increased from 44% in 2007 to 49% in 2012 and to 57% in 2017 [12]. Based on UNICEF data on infant and young child feeding, the prevalence of EIBF in Indonesia is still low compared to that in other countries of South-East Asia such as Cambodia (63%), Myanmar (67%) and Timor-Leste (75%) and almost equal to that in the Philippines (56.9%), although it is higher than that in Vietnam (27%), Thailand (40%) and Lao People's Democratic Republic (50%) [34]. The WHO and UNICEF guidelines on breastfeeding classify the percentage of babies initiated into breastfeeding in the first hour of birth as Poor if EIBF is less than 30%, Fair if EIBF is between 30% and 49%, Good if EIBF is between 50% and 85% and Very Good if EIBF is between 90% and 100% [35]. Based on these guidelines, the practice of EIBF in Indonesia may be considered "Good" as 57% of the babies are given mother's breastmilk within one hour of their birth. The Government of Indonesia has, of long been supporting programs to improve maternal and child health in the country, including improvements in the prevalence of early and overall breastfeeding. Article 9 of the Indonesian *Government Regulation Number 33 of 2012* states that health providers and health facilities are mandated with promoting the initiation of breastfeeding within one hour of the birth of a child. They are also expected to implement and support exclusive breastfeeding guided by the *Ten Steps to Successful Breastfeeding*, including ensuring that their staff have sufficient knowledge, competence and skills to support breastfeeding, facilitate immediate and uninterrupted skin-to-skin contact and support mothers to initiate breastfeeding as soon as

possible after delivery [13,14]. However, as of 2017 more than 43% of newborn babies in Indonesia are not initiated into breastfeeding within the first hour of birth.

Unlike some previous studies which have shown that mother's age is associated with EIBF [14,15,36], the present study shows no statistically significant association between mother's age and EIBF. This exception may be attributed to the difference between the age-distribution of present the study sample and that of the studies referred to above. The present study sample consists of less than 5% of the women younger than 20 years and only 6% older than 40 years.

The present study has also shown that there is no association between EIBF and parent's education or rural/urban residence. The association between EIBF and mother's rural/urban residence is uncertain. While some studies have shown that women in rural areas have higher odds of EIBF [13,16,19], some other studies have shown that urban women have higher odds of EIBF [11,17,18]. In terms of education however, studies in Ethiopia and Malawi have shown that higher maternal education is associated with higher odds of EIBF [11,13]. This apparent difference between the findings of the studies in Ethiopia and Malawi and that of the present study may be explained by the fact that in the present study, EIBF varies very little according to both mother's education and father's education.

Birth order of the child is also a significant predictor of EIBF, with higher order births being more likely to be initiated into early breastfeeding compared to first order births. This is consistent with the findings of studies which show that multiparous women (i.e., women delivering second or higher order births) are more likely to practice EIBF than primiparous women (i.e., women giving first order births) [14–16]. One plausible explanation for this is that women with only one child may be very young and have little knowledge about infant feeding, while women with more than one child may be older and they become experienced in breastfeeding from their previous pregnancies, so they become better at practising EIBF as they have more children [13]. Some studies have also noted that previous experience of breastfeeding has a positive impact on the intention and timeliness of future breastfeeding because such experience introduces positive changes in beliefs in breastfeeding [17,18]. Unexpectedly, mothers from wealthier households have lower odds of EIBF compared to mothers from poorer households. This is unexpected because women living in wealthier households are generally more educated, and higher education promotes better knowledge about breastfeeding and health [36]. A study in Nigeria showed that wealthier women are more likely to practise EIBF compared to poorer women [18]. However, women in the present study may be exposed to modernisation which would probably influence them to use substitutes of breastmilk [24]. On the other hand, women living in households with poor wealth index may have financial limitations, therefore they practise EIBF better than their richer counterparts because breastmilk is "free" [24,37].

Compared to unemployed women, working women are less likely to breastfeed early. This may be due to the possibility that working women might be concerned about their professional and body image, so they might consider breastfeeding an obstacle to return to their physical shape of the time before they became pregnant [18,24]. Further, working women have access to income and therefore they may also afford to have deliveries by C-section and access supplementary infant food such as infant formula, both of which are associated with DIBF [38]. Therefore, it is very important to provide lactation counselling to women, especially women who are working in order to assist them address their breastfeeding problems and practise EIBF.

Women whose partners work in non-agricultural activities are found to be more likely to breastfeed early compared to women whose partners are not working. Father's support is important for improving EIBF [39–41]. Paternity leave has positive influence on breastfeeding because fathers are able to take care of their wives and children [40]. Paternal support on

breastfeeding practices include verbal encouragement, partner's responsiveness to the needs of mothers, father's assistance in managing and solving initial breastfeeding difficulties and performing household chores and looking after the older children [39,40]. However, children of non-working fathers may have working mothers who are found to be less likely to practice EIBF. Therefore, it is important to improve paternal involvement in breastfeeding by equipping the fathers with information about the benefits of breastfeeding, ways of overcoming breastfeeding problems and by providing them with the needed skills and enable them to support and help their wives through breastfeeding difficulties.

As expected, women who have early skin-to-skin contact with their babies are more likely to initiate breastfeeding early. The adjusted odds ratio of this variable (adjusted OR = 2.62) is the highest among the OR for all other variables implying that skin-to-skin contact is the strongest predictor of EIBF in Indonesia 2017. This influence of skin-to-skin contact is in line with evidence from Japan and Zimbabwe which shows that skin-to-skin contact can improve EIBF, improve mother-baby relationship and improve maternal satisfaction and confidence with the first breastfeeding, all of which are crucial for the survival of the newborn child [16,21]. Further, skin-to-skin contact is free of cost, easy and beneficial to both mothers (as mentioned above) and their infants with stabilising their body temperature, and preventing suckling/breastfeeding difficulties [42–44]. It is important to improve skin-to-skin contact, especially after a C-section, be it in the operating room and/or during recovery. Health providers such as doctors and nurses should play an important role during routine care after a C-section in advocating and assisting mothers to practice skin-to-skin contact and EIBF.

In terms of size at birth, babies born with average and large sizes are more likely to be initiated into early breastfeeding than babies with small sizes at birth. This finding is similar to those from studies in Nepal [38] and West Africa [37]. Mothers and healthcare providers perceive that large size infants are healthy and they are associated with physical maturity, proper functions of suckling and deglutination [37,38]. Therefore, average or large sizes at birth may facilitate EIBF. Conversely, small sized babies at birth are perceived as being physically immature, unable to suckle properly and have difficulty in swallowing [37,38,45]. Additionally, small sized or low birth weight babies tend to suffer from morbidities and require intervention with separation from their mothers for longer periods after delivery, resulting in delays in the initiation of breastfeeding [16,18]. Therefore, it is important for birth attendants to promote, encourage, assist and support EIBF, especially among mothers with small sized babies.

The findings of the present study also show that women who deliver with a C-section are less likely to initiate early breastfeeding. This is consistent with similar findings in other countries [11,16–22]. A C-section results in prolonged maternal-infant separation, maternal endocrinological changes induced by the surgery and stressful conditions for the woman, who would also need time to recover from the effects of anaesthesia [11,20,46]. Moreover, infants born by a C-section might develop respiratory distress for which they could be taken to newborn intensive care units [11]. Therefore, it is recommended that women with C-section deliveries should receive special counselling in breastfeeding and support from trained providers about how to breastfeed properly. Further, providing adequate support and guidance from well-trained health workers in the early postnatal period may improve EIBF among mothers delivering with C-sections [47,48].

The findings of the present study further indicate that mothers who deliver their babies with the assistance of skilled birth attendants are more likely to practise EIBF. These results are similar to those of other studies such as qualitative research on barriers and facilitators in Ghana in 2008 [23], secondary analysis of EIBF based on the 2016 Ethiopian Demographic and Health Survey [17], and the secondary analysis of EIBF based on the 2015 Zimbabwe Demographic and Health Data [16]. The provision of breastfeeding support from health

workers is important. Skilled Birth Attendants have knowledge about the benefits of EIBF including the benefit that EIBF facilitates the delivery of placenta, prevents postpartum bleeding, helps with infant-mother bonding and facilitates lactation [23]. Because of their knowledge and support, many health workers are successful in teaching the mothers to practise EIBF. Successful breastfeeding by mothers is also related to satisfaction, so mothers and their babies can enjoy the breastfeeding process and feel relaxed [21,49]. Therefore, it is important to provide guidelines, training manuals, and counselling at health facilities to improve women's perceptions of breastfeeding initiation.

This study has several implications. EIBF followed by continued exclusive breastfeeding for the first six months of a baby's life and breastfeeding with complementary feeding for up to two years and beyond have been parts of WHO/UNICEF recommendations [5]. It is important to promote and support EIBF in health facilities which provide maternity and newborn services. EIBF is found to be associated with the reduction of neonatal mortality by increasing the rates of exclusive breastfeeding [6,9]. Besides strengthening health providers' competence in lactation management and infant feeding, it is important to regularly update their knowledge on how to practice EIBF properly. Several initiatives from other countries can be adapted in Indonesian conditions in order to increase breastfeeding practices. These initiatives include awareness raising campaigns among communities leaders, and engaging stakeholder to incorporate breastfeeding in national or local nutritional policies, referral pathway and support networks [20,47].

The strength of the present study is that the 2017 IDHS data are nationally representative so that the recommendation arising out of this research can be implemented in all the provinces of Indonesia. This study, however, has some limitations. The study is based on information given by women about breastfeeding their most recent child born in the last 24 months preceding the survey and therefore, as a retrospective information it may have errors due to recall lapse. Another limitation is the cross-sectional design of the sample that cannot describe any cause and effect relationship. Moreover, since Indonesia is culturally and geographically diverse, further studies are needed to identify the predictors of EIBF in specific cultural and geographical contexts so that the practice of EIBF can be improved throughout the country.

## Conclusion

About six out of ten newborn babies in Indonesia have EIBF. The strongest predictor of EIBF is skin-to-skin contact between mother and infant immediately after birth, implying that this determinant is key to successful of EIBF. Skin-to-skin contact is one of the ten steps to successful breastfeeding. It is an easy and inexpensive way to improve maternal breastfeeding self-efficacy and confidence for which it is needed to provide experienced and trained lactation consultants to assist the mothers with early breastfeeding practice after delivery. It is also important to focus on designing policies to promote EIBF, especially among mothers with non-vaginal deliveries such as a C-section. The findings of the study also point out the importance of improving awareness of women, their spouses and birth attendants about the significant benefits of EIBF. All these interventions would substantially help the government in continuously promoting and supporting early initiation of breastfeeding and its continuation for up to two years or beyond for the lifelong health of newborn children and their mothers.

## Acknowledgments

This study used the dataset from the 2017 Indonesia Demographic and Health Survey. The authors would like to thank to DHS Program for giving the permission to use the IDHS 2017

data for this analysis. The authors would like to acknowledge National Population and Family Planning Board for the support in scientific writing.

## Author Contributions

**Conceptualization:** Maria Gayatri.

**Formal analysis:** Maria Gayatri.

**Methodology:** Gouranga Lal Dasvarma.

**Supervision:** Gouranga Lal Dasvarma.

**Validation:** Gouranga Lal Dasvarma.

**Writing – original draft:** Maria Gayatri.

**Writing – review & editing:** Maria Gayatri, Gouranga Lal Dasvarma.

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
