## [Decision Letter · Decision Letter 0]

1 Apr 2020

PONE-D-20-03622

Predictors of early initiation of breastfeeding in Indonesia: A population-based cross-sectional survey

PLOS ONE

Dear Mrs. Gayatri,

Thank you for submitting your manuscript to PLOS ONE. After careful consideration, we feel that it has merit but does not fully meet PLOS ONE’s publication criteria as it currently stands. Therefore, we invite you to submit a revised version of the manuscript that addresses the points raised during the review process. Please follow all reviewers´ recommendations, with special attention to English editing.

We would appreciate receiving your revised manuscript by May 16 2020 11:59PM. To enhance the reproducibility of your results, we recommend that if applicable you deposit your laboratory protocols in protocols.io, where a protocol can be assigned its own identifier (DOI) such that it can be cited independently in the future. For instructions see: http://journals.plos.org/plosone/s/submission-guidelines#loc-laboratory-protocols

We look forward to receiving your revised manuscript.

Kind regards,

Marly A. Cardoso, Ph.D.

Academic Editor

PLOS ONE

Journal Requirements:

Reviewers' comments:

Reviewer's Responses to Questions

**Comments to the Author**

1. Is the manuscript technically sound, and do the data support the conclusions?

Reviewer #1: Yes

Reviewer #2: Yes

Reviewer #3: Partly

2. Has the statistical analysis been performed appropriately and rigorously? 

Reviewer #1: I Don't Know

Reviewer #2: Yes

Reviewer #3: Yes

3. Have the authors made all data underlying the findings in their manuscript fully available?

Reviewer #1: Yes

Reviewer #2: Yes

Reviewer #3: Yes

4. Is the manuscript presented in an intelligible fashion and written in standard English?

Reviewer #1: No

Reviewer #2: No

Reviewer #3: Yes

5. Review Comments to the Author

Reviewer #1: The manuscript submitted by Maria Gayatri and colleagues aims to investigate predictors of early initiation of breastfeeding in Indonesia. Overall, this manuscript presents results that would be of interest to the community of scientists and clinicians concerned with this problem. Prior to publication, the following points should be addressed.

Overall: I recommend editing the text to achieve a more appropriate scientific language. Furthermore, the quality of writing should be verified, since punctuation marks issues can be identified over the manuscript, compromising the structure and demanding extra attention to interpret the information.

Abstract:

-Introduction and methods repeat the objective of the study. Please, mention it just once.

-You mentioned “EIBF”, but this term was fully described later.

-Please, add the confidence Interval for EIBF proportion.

Introduction:

-The introduction is very succinct. Authors could add information about the factors that most commonly affect EIBF and enrich the text with data on the global epidemiology of EIBF, since these are the objectives of the study (% of EIBF and predictors).

-You can start using “EIBF” abbreviation in this section.

Methods:

Dependent variable:

-For definition of EIBF you can also cite the “Ten Steps to Successful Breastfeeding” of WHO.

-I recommend you reorganize the text as follow: mention the dependent variable, its definition and finally how it was measured and categorized in the analysis.

Independent variable:

-The wealth index of the woman’s family is not a medical characteristic. Please, explain how this variable was constructed and what it was based on.

-The text is not well organized. The paragraph between line 100 to 106 - in page 4, is repeated in the text.

-The levels of the independent variable are not clear. Please, reorganize the text explaining the levels, the variables included in each level, and how they were categorized for analyses.

-The following variables need more details for better understanding: type of assistance during delivery and child’s size at birth (in lieu of birth weight).

Methods of Analysis:

-I cannot comment on complex samples design statistical method as I am not familiar with it.

-Please, add the abbreviation of odds ratio in brackets.

-Bernoulli's formula was presented but not mentioned in the text as “formula (1)”.

-what program was used to perform the statistical analyses?

-what does “(n.d.)” mean? Page 6, line 145.

Results:

-You mentioned “early initiation of breastfeeding (EIBF)” before, so use the acronym from then on. Make sure you align this detail throughout all the text.

-The outcome of interest was previously defined. Please, do not repeat the information in this section.

-What table was the following affirmation based on? “The distributions of early initiators of breastfeeding….with respect to each of the predictor variables appear not to be dissimilar from one another…” Table 1 does not allow this affirmation.

Table 1:

-Mother's age at childbirth has n=6.615, please mention in footnotes if there are any missing data.

-please, note that some variables have n=6.617 instead of the referred 6.616 (Birth order of the child, Skin-to-skin contact, etc.). Check all variables and inform the correct n.

-information on “education and working status of the parents” is confused. Please, review the writing (line 182 to 191). For instance, with respect to the following sentence: “However, the opposite appears to be true with respect to father’s education, where more children of nonworking fathers experience delayed initiation of breastfeeding.”, are you talking about father’s education or working status?

-With respect to the following sentences: “Perhaps children whose fathers are not working have working mothers…”, “….it should be because the importance of early initiation of breastfeeding must be conveyed to the pregnant women during their ANC visits”, “C-section deliveries may have higher levels of education as found elsewhere [19],” This kind of comments/suggestions is appropriate in the discussion section. Please, try to be more objective in the result section by presenting only the interpretation of the results shown in tables. See the following example: (mother living with her partner and having one or more children were positively associated with a higher probability of EBF. Conversely, mother aged <19 years and using pacifier, were related to a lower probability of EBF. However, there were no significant associations between EBF and other variables, including maternal education, self-reported skin color and household wealth index).

-About “Place of delivery (home versus health facility)”, should be described in the methods section.

-Line 210, “…as indicated by the χ2 and p-values”, the statistical analysis was described before. Avoid repeating information in the text.

-“(Table 2) in line 210”. You mentioned before that you were talking about this table.

Title: Predictors of Early Initiation Breastfeeding or Predictors of Early Initiation of Breastfeeding?

-The first paragraph after this title is not necessary (line 221 to 234). You have already defined the outcome of interest, explained the statistical analysis and described the factors associated in the bivariate analyses.

-Details on “father’s work (or occupation of the woman’s husband)” categorization (in agricultural and non-agricultural activities) was not given in methods.

-Is the fact father works in agricultural activities related to EIBF? (OR 1.85 (0.94, 3.65). Please, review the interpretation of the results.

-Table 3 was also calculated by the authors from 2017 IDHS data?

-Table 3 does not need to show the p-value.

-I recommend presenting the results in the same order in which the variables were listed in the tables (according to each level of variables).

Discussion

-EIBF prevalence is higher than that in the Philippines (56.9%)?

-If “health providers and health facilities are mandated with promoting the initiation of breastfeeding within one hour of the birth of a child”, why is the prevalence of EIBF in Indonesia still lower than some other countries? According to WHO, a country is considered “very Good” when at least 90% of the children initiate BF within one hour of life. Indonesia would be considered “Good”. I recommend you cite and discuss these ratings. (WHO reference: Infant and Young Child Feeding. A tool for assessing national practices, policies and programmes. Geneva. 2003)

-About skilled birth attendants: “The results of the present study are similar to those of other studies”, where were these studies carried out? in what year were they performed? please give more details of the studies you are comparing your results with.

-Please compare the results of "size at birth" with other studies. This result has not been widely discussed.

-(line 332) “mothers from middle households had lower odds of EIBF compared to those from poorer households”, but it was not significant, so why are you mentioning this category?

-(line 334) “women living in households with middle and rich wealth index are supposed to be more educated and more knowledgeable about the benefits of EIBF”. Please, cite a reference for this affirmation. Please, compare the results with other studies besides reference 28.

- Mother working status result has not been widely discussed.

-I recommend discussing the results in the same order as they were presented in the results section.

Reviewer #2: This study examines the predictors of early initiation of breastfeeding among Indonesian women. It stands out for the use of a large and nationally representative sample; however, certain considerations need to be made:

The paper could be written more concisely. Some parts are repetitive and confusing (for example, lines 98-109) and some sentences are unnecessary (for example, lines 221-228). I recommend that the abstract should be rewritten to make it clearer and objective. In the methods section, authors should include more information about data collection (for example, how was the questionnaire applied?) and rewrite the description of covariates to make it clearer and easier to follow. The results section should contain only results, no discussion. This section can be more concise as well. Table 1 and 2 are very similar, so the authors should consider presenting a combined table. Lastly, in the conclusion section, I suggest including a summary of the main findings of this study (which are the main predictors associated with the outcome).

I do have some minor comments below:

1. Line 145: Please check the information inside the parentheses.

2. Line 146: What does ICF mean?

3. Line: The title of Table 1 can be improved.

Reviewer #3: Overall comment: This is a paper exploring the predictors of early initiation of breastfeeding using DHS data from Indonesia. This is a topic with a strong body of evidence with data report from multiple countries. The paper has merit and brings reliable analysis, however, the introduction needs to be reviewed including detail about the context (Indonesia) as well as more detail should be provided in the methods section and extensive revisions in reporting results are needed. Below I provide a detail comment and suggestions for the authors to improve the manuscript.

ABSTRACT

Lines 13-15. Please clearly define your research question and its importance. The introduction section can be expanded to briefly contextualize why it is important to understand the predictors of early breastfeeding initiation in general and why to study this problem is important to the given population.

Lines 16-24. In the method section, the first sentence can be removed as it repeats the objective of the study already stated.

Lines 24-28. Revise the report of results including OR and comparison groups when appropriate.

Line 29-32. Revise the conclusion to reply to your research question(s). If the focus of the study is breastfeeding within the first hour why its conclusion is addressing breastfeeding for up to years, please keep your conclusion limited to what your data can support.

INTRODUCTION

Overall comment. I would recommend expanding the introduction to state the problem beyond the benefits of breastfeeding. The authors should consider including one or two paragraphs in the introduction demonstrating what has been identified in prior studies and what are the gaps. Why is important to identify predictors of early breastfeeding initiation? What is this studying adding to the literature or to the context? Early breastfeeding initiation is influenced by several modifiable factors - there is a body of evidence showing factors associated with this practice and none were mentioned. Also, early breastfeeding initiation is highly linked with hospital practices, for instance, this is a practice endorsed and part of the Baby-Friendly Hospital Initiative which was not mentioned in the introduction.

Line 57-58. Please clarify what means “losing their child in the neonatal period”. Is this related to increased neonatal mortality? If so, provide the rationale and more evidence on this statement

METHODS

Overall comment:

The methodological steps described in this section does not match with the results presented. Please revise the methodological steps: 1) descriptive analysis, 2) bivariate analysis - indicating which statistics were used (chi-square or logistic regression, not both), 3) multivariate analysis (predictors of EIBF) and make clear the goal of each of methodological steps. For each step, it is expected a matching table displacing results.

The authors conducted a bivariate analysis to select predictors of breastfeeding early initiation and this name should be used throughout instead of “simple” logistic regression. Please mention the level of significance that what was considered in the bivariate analysis to select variables to the multivariate analysis. Also, please clarify why was used a chi-square test and bivariate logistic regression and how both analyses informed the selection of independent variables. Please provide a rationale to use both methods since this is not clear in the methods or results section.

Please review the English verbal tense throughout. A few parts of the methodology are in present and others present perfect, please choose one and be consistent throughout.

Minor comments:

Line 84-91. Please clarify if mothers who did not breastfeed were included or excluded.

Line 92-109. Why were these predictors chosen? Was this based on literature review or data availability? What is the plausibility of selecting these predictors? Was there any framework used to select variables?

Line 110. Remove the word “methods”

Line 111. Remove the word “basically”

Line 114 -115. Remove the word “simple” on both lines

Line 115. Remove the word “simple” logistic regression.

Results

Overall comment:

The result section is long and repetitive. It has to be streamlined with the description of methodological steps. Tables 1 or 2 should be removed or revised to show a descriptive analysis of the whole population (not per outcome) and bivariate analysis. In the text, authors should focus their report in the most important results not show interpretations, interpretations should be placed in the discussion section. For instance, the authors could highlight important characteristics of the population studied and which were the variables selected in the bivariate to the multivariate analysis.

Line 160 - Remove the word the definition for early breastfeeding initiation. Once this was defined in the method section there is no need for repetition. Also, if authors will opt for using an acronym for early initiation of breastfeeding (EIBF), please define it for the first time in the text and use this throughout.

Tables 1 and 2. Please clarify and justify the need for having these two tables. I strongly recommend removing table 1 and keep with table 2.

Line 156. Please remove subheading “Characteristics of respondents”

Line 220. Please remove subheading “Predictors of Early Initiation Breastfeeding”

Lines 161-210. Please revise accordingly overall comment above.

Lines 221-228. Consider summarizing it and moving it to the analysis section.

Lines 235-261. Please summarize and revise accordingly the overall comment above.

Table 3 is confusing, please make sure it is necessary to present bivariate analysis results in this table. Again, consider choosing between chi-square and bivariate logistic regression to select variables to multivariate model, detail this step in the methods, and provide a clear cut-off point to include or not a variable in the model.

Minor comment: please make sure tables follow the same formatting style.

Discussion:

Overall comment: Consider moving a few sentences from Indonesia context to the introduction, this will help readers to understand the context of the study earlier. Also, was there any hypothesis the authors were trying to explore? Or was this an exploratory study? What gaps this analysis fills?

Lines 285-288/298-300/310-312. Do not report results in the discussion.

Line 316 – this sentence seems to be incomplete.

Conclusion: The conclusion lacks to respond to the research question. Why is important to know the predictors of EIBF?

6. PLOS authors have the option to publish the peer review history of their article (what does this mean?). If published, this will include your full peer review and any attached files.

Reviewer #1: No

Reviewer #2: No

Reviewer #3: No

---

## [Author Response · Author response to Decision Letter 0]

13 May 2020

Reviewer #1: The manuscript submitted by Maria Gayatri and colleagues aims to investigate predictors of early initiation of breastfeeding in Indonesia. Overall, this manuscript presents results that would be of interest to the community of scientists and clinicians concerned with this problem. Prior to publication, the following points should be addressed.

Overall: I recommend editing the text to achieve a more appropriate scientific language. Furthermore, the quality of writing should be verified, since punctuation marks issues can be identified over the manuscript, compromising the structure and demanding extra attention to interpret the information.

Response: We have reformatted the manuscript to a more appropriate scientific language. 

Abstract:

-Introduction and methods repeat the objective of the study. Please, mention it just once.

-You mentioned “EIBF”, but this term was fully described later.

-Please, add the confidence Interval for EIBF proportion.

Response: Thank you. We have added the confidence interval for EIBF proportion. We have revised the abstract and only mentioned the objective just once. 

Introduction:

-The introduction is very succinct. Authors could add information about the factors that most commonly affect EIBF and enrich the text with data on the global epidemiology of EIBF, since these are the objectives of the study (% of EIBF and predictors).

-You can start using “EIBF” abbreviation in this section.

Response: Thank you for your comments. We have added data on global epidemiology of EIBF and discussed the factors influencing EIBF. We have started using the acronym EIBF in this section after mentioning early Initiation of Breastfeeding with expanded initials at the beginning of the Introduction. We have also mentioned The Ten Step to Successful Breastfeeding from WHO/UNICEF with references. 

Methods:

Dependent variable:

-For definition of EIBF you can also cite the “Ten Steps to Successful Breastfeeding” of WHO.

Response: Thank you. We have added the definition of EIBF based on the WHO/UNICEF initiative - Then Steps to Successful Breastfeeding. 

-I recommend you reorganize the text as follow: mention the dependent variable, its definition and finally how it was measured and categorized in the analysis.

Response: For the dependent variable we have added and reorganised as suggested by the reviewer. 

Independent variable:

-The wealth index of the woman’s family is not a medical characteristic. Please, explain how this variable was constructed and what it was based on.

Response: Thank you for your suggestion. We have described Wealth Index as a household level variable. This variable was constructed by The Demographic and Health Survey team, but we have added the details of how this index was constricted with adequate references. 

-The text is not well organized. The paragraph between line 100 to 106 - in page 4, is repeated in the text

Response: Thank you. We have deleted the repeated text in the methods section. 

-The levels of the independent variable are not clear. Please, reorganize the text explaining the levels, the variables included in each level, and how they were categorized for analyses.

Response: Thank you for pointing this out. The method has been corrected and organized according to individual level, household level and community level variables. The article has been corrected on line 120-162 (check the line in the final manuscript).

-The following variables need more details for better understanding: type of assistance during delivery and child’s size at birth (in lieu of birth weight).

Response: We have added in the methods section the details of type of assistance during delivery and child’s size at birth. 

Methods of Analysis:

-I cannot comment on complex samples design statistical method as I am not familiar with it.

-Please, add the abbreviation of odds ratio in brackets.

Response: Thank you. We added the abbreviation of adds ratio in the bracket. 

-Bernoulli's formula was presented but not mentioned in the text as “formula (1)”.

Response: Noted. We have added “formula (1)” in the text. Thank you.

-what program was used to perform the statistical analyses?

Response: We have used Stata version 15 (StataCorp, College Station, TX). We have added the statistics tools in the text. 

-what does “(n.d.)” mean? Page 6, line 145.

Response:” n.d” stands for “not dated”. This is a standard notation in citing a reference for which the date or year of publication is not available. However, we have deleted “n.d.” in the text.

Results:

-You mentioned “early initiation of breastfeeding (EIBF)” before, so use the acronym from then on. Make sure you align this detail throughout all the text.

Response: Thank you. We have revised the paper accordingly and by mentioning Early Initiation of Breastfeeding with expanded initials at the beginning and used the acronym EIBF in the remainder of the text. 

-The outcome of interest was previously defined. Please, do not repeat the information in this section.

Response: Thank you, we have deleted the definition of outcome in the result because it is already mentioned in the methods section. 

-What table was the following affirmation based on? “The distributions of early initiators of breastfeeding….with respect to each of the predictor variables appear not to be dissimilar from one another…” Table 1 does not allow this affirmation.

Response: Thank you. We have deleted this sentence since it does not have much bearing on the results of the analysis. 

Table 1:

-Mother's age at childbirth has n=6.615, please mention in footnotes if there are any missing data.

-please, note that some variables have n=6.617 instead of the referred 6.616 (Birth order of the child, Skin-to-skin contact, etc.). Check all variables and inform the correct n.

Response: Thank you. There are no missing data. We have corrected the error. Now the n is 6,617 throughout. However, we have removed Table 1 because it is very nearly similar to Table 2. Now there is only table showing the distribution of respondents and their characteristics. 

-information on “education and working status of the parents” is confused. Please, review the writing (line 182 to 191). For instance, with respect to the following sentence: “However, the opposite appears to be true with respect to father’s education, where more children of nonworking fathers experience delayed initiation of breastfeeding.”, are you talking about father’s education or working status?

Response: Thank you. We are referring to father’s occupation. We have revised the text to make this clear. 

-With respect to the following sentences: “Perhaps children whose fathers are not working have working mothers…”, “….it should be because the importance of early initiation of breastfeeding must be conveyed to the pregnant women during their ANC visits”, “C-section deliveries may have higher levels of education as found elsewhere [19],” This kind of comments/suggestions is appropriate in the discussion section. Please, try to be more objective in the result section by presenting only the interpretation of the results shown in tables. See the following example: (mother living with her partner and having one or more children were positively associated with a higher probability of EBF. Conversely, mother aged <19 years and using pacifier, were related to a lower probability of EBF. However, there were no significant associations between EBF and other variables, including maternal education, self-reported skin color and household wealth index).

Response: We have moved these comments to the discussion section. Regarding the example you have given, may we know its source?

-About “Place of delivery (home versus health facility)”, should be described in the methods section.

Response: We have added the variable of place of delivery in the methods section. 

-Line 210, “…as indicated by the χ2 and p-values”, the statistical analysis was described before. Avoid repeating information in the text.

Response: Thank you. We have deleted the repetition.

-“(Table 2) in line 210”. You mentioned before that you were talking about this table.

Title: Predictors of Early Initiation Breastfeeding or Predictors of Early Initiation of Breastfeeding?

Response: The previous Table 2 has changed to be Table 1 in the revised manuscript. We have revised the title of the (new) Table 1 into “Characteristics of women having their most recent birth in the two years preceding the survey according to initiation of breastfeeding status, Indonesia 2017”. ” 

-The first paragraph after this title is not necessary (line 221 to 234). You have already defined the outcome of interest, explained the statistical analysis and described the factors associated in the bivariate analyses.

Response: Agreed. We have removed the statements.

-Details on “father’s work (or occupation of the woman’s husband)” categorization (in agricultural and non-agricultural activities) was not given in methods.

Response: We have added it in the methods section. 

-Is the fact father works in agricultural activities related to EIBF? (OR 1.85 (0.94, 3.65). Please, review the interpretation of the results.

Response: We have revised the article. There is no difference in the practicing of EIBF between children whose fathers are not working or working in agriculture. However, children whose fathers are working in non-agriculture activities are about twice as likely to have an EIBF compared to children whose fathers are not working. 

-Table 3 was also calculated by the authors from 2017 IDHS data?

Response: Yes, table 3 was calculated by the authors from 2017 IDHS data. In the revised manuscript, Table 3 becomes Table 2. Thank you. 

-Table 3 does not need to show the p-value.

Response: We thank the reviewer for this comments. However, we believe that p-value is important to conclude the analysis. Therefore, we have retained the p-values in Table 2. 

-I recommend presenting the results in the same order in which the variables were listed in the tables (according to each level of variables).

Response: We appreciate the reviewer’s comment and agree that presenting the results in the same order is important. We have revised it for improvement to our paper. 

Discussion

-EIBF prevalence is higher than that in the Philippines (56.9%)?

Response: Thank you for pointing this out. EIBF prevalence in Indonesia is 56.6% and it is almost equal to that in the Philippines (56.9%). Both round up to 57%.

-If “health providers and health facilities are mandated with promoting the initiation of breastfeeding within one hour of the birth of a child”, why is the prevalence of EIBF in Indonesia still lower than some other countries? According to WHO, a country is considered “very Good” when at least 90% of the children initiate BF within one hour of life. Indonesia would be considered “Good”. I recommend you cite and discuss these ratings. (WHO reference: Infant and Young Child Feeding. A tool for assessing national practices, policies and programmes. Geneva. 2003)

Response: Thank you for pointing out this useful source. We have added the WHO reference that based on the guidelines for writing, the practice of EIBF in Indonesia is categorized as good (50-85%). 

-About skilled birth attendants: “The results of the present study are similar to those of other studies”, where were these studies carried out? in what year were they performed? please give more details of the studies you are comparing your results with.

Response: We have added the studies and the year to the manuscript then compared with our results. 

-Please compare the results of "size at birth" with other studies. This result has not been widely discussed.

Response: We have added baby’s size at birth to the discussion and put the justification based on other studies about EIBF. 

-(line 332) “mothers from middle households had lower odds of EIBF compared to those from poorer households”, but it was not significant, so why are you mentioning this category?

Response: Agreed. We have removed it.

-(line 334) “women living in households with middle and rich wealth index are supposed to be more educated and more knowledgeable about the benefits of EIBF”. Please, cite a reference for this affirmation. Please, compare the results with other studies besides reference 28.

Response: We have provided evidence that women living in households with middle and rich wealth index are more educated and that higher education promotes better knowledge about breastfeeding and health. 

- Mother working status result has not been widely discussed.

Response: We have added related to mother working status under the discussion of wealthier households.

-I recommend discussing the results in the same order as they were presented in the results section.

Response: Thank you. We have revised the discussion in the same order as in the methods and results section. 

Reviewer #2: This study examines the predictors of early initiation of breastfeeding among Indonesian women. It stands out for the use of a large and nationally representative sample; however, certain considerations need to be made:

The paper could be written more concisely. Some parts are repetitive and confusing (for example, lines 98-109) and some sentences are unnecessary (for example, lines 221-228). 

Response: We have revised the paper and removed the repetitions as far as possible. 

I recommend that the abstract should be rewritten to make it clearer and objective. 

Response: We have revised the abstract.

In the methods section, authors should include more information about data collection (for example, how was the questionnaire applied?) and rewrite the description of covariates to make it clearer and easier to follow. 

Response: Thank you for your comments. We have re-written the methods in the same order as that of the order of analysis to clearer and easier to understand. We have also described the data collection in more detail. 

The results section should contain only results, no discussion.

Response: Thank you. We have now included only the results here and moved the rest to the Discussion section. 

 This section can be more concise as well. Table 1 and 2 are very similar, so the authors should consider presenting a combined table. 

Response: Thank you for your suggestion, we have removed Table 1 and kept Table 2 (which has now become Table 1). In the revised manuscript, we only use (the new) Table 1 to show the characteristics of the respondents according to early breastfeeding status. . 

Lastly, in the conclusion section, I suggest including a summary of the main findings of this study (which are the main predictors associated with the outcome).

Response: Thank you. We have added a summary of the main findings. 

I do have some minor comments below:

1. Line 145: Please check the information inside the parentheses.

Response: Thank you. We have removed the information inside the parentheses. (n.d.) means not dated, i.e. the date or year of publication of the source is not known. 

2. Line 146: What does ICF mean?

Response: According to Wikipedia (https://en.wikipedia.org/wiki/ICF_International):

 “ICF was founded in 1969 as the Inner City Fund. …. The company was reorganised as a consulting firm and renamed itself ICF Incorporated. In 2006, the consultancy was renamed ICF International….” .

However, in all publications and references it is cited as ICF International, which provides technical assistance to all Demographic and Health Surveys (DHS) Program, which is funded by the U.S. Agency for International Development (USAID). ICF International assisted in the preparation of the 2017 Indonesia Demographic and Health Survey report writing. 

3. Line: The title of Table 1 can be improved.

Response: Thank you. We have deleted Table 1 because it is very nearly similar to Table 2. 

Reviewer #3: Overall comment: This is a paper exploring the predictors of early initiation of breastfeeding using DHS data from Indonesia. This is a topic with a strong body of evidence with data report from multiple countries. The paper has merit and brings reliable analysis, however, the introduction needs to be reviewed including detail about the context (Indonesia) as well as more detail should be provided in the methods section and extensive revisions in reporting results are needed. Below I provide a detail comment and suggestions for the authors to improve the manuscript.

Response: We have revised the Introduction, and the methods and results sections to address your comments.

ABSTRACT

Lines 13-15. Please clearly define your research question and its importance. The introduction section can be expanded to briefly contextualize why it is important to understand the predictors of early breastfeeding initiation in general and why to study this problem is important to the given population.

Response: Thank you. We have revised the Abstract to address your comment.

Lines 16-24. In the method section, the first sentence can be removed as it repeats the objective of the study already stated.

Response: Thank you. We have removed the first sentence.

Lines 24-28. Revise the report of results including OR and comparison groups when appropriate.

Response: We have added the OR and comparison groups in the results.

Line 29-32. Revise the conclusion to reply to your research question(s). If the focus of the study is breastfeeding within the first hour why its conclusion is addressing breastfeeding for up to years, please keep your conclusion limited to what your data can support.

Response: We have revised the Conclusion to address your comment.

INTRODUCTION

Overall comment. I would recommend expanding the introduction to state the problem beyond the benefits of breastfeeding. The authors should consider including one or two paragraphs in the introduction demonstrating what has been identified in prior studies and what are the gaps. Why is important to identify predictors of early breastfeeding initiation? What is this studying adding to the literature or to the context? Early breastfeeding initiation is influenced by several modifiable factors - there is a body of evidence showing factors associated with this practice and none were mentioned. Also, early breastfeeding initiation is highly linked with hospital practices, for instance, this is a practice endorsed and part of the Baby-Friendly Hospital Initiative which was not mentioned in the introduction.

Line 57-58. Please clarify what means “losing their child in the neonatal period”. Is this related to increased neonatal mortality? If so, provide the rationale and more evidence on this statement

Response: Losing a child in the neonatal period means the death of the child in the neonatal period. Sources supporting this statement have been cited in the sentence. The rationale why a child might die in the neonatal period when breastfeeding is initiated late is that the child does not receive “the highly nutritious maternal colostrum that reduces the risk of microbial translocation, accelerates intestinal maturation and promotes resistance to infection [5–9]”. This is stated in the previous sentences. 

METHODS

Overall comment:

The methodological steps described in this section does not match with the results presented. Please revise the methodological steps: 1) descriptive analysis, 2) bivariate analysis - indicating which statistics were used (chi-square or logistic regression, not both), 3) multivariate analysis (predictors of EIBF) and make clear the goal of each of methodological steps. For each step, it is expected a matching table displacing results.

Response: The results are presented in the order suggested by the reviewer. There are only two tables included in the revised manuscript. We have removed the chi-square results and used only the bivariate logistic regression for bivariate analysis. Table 2 refers respectively to bivariate analysis and logistic regression (multivariate) analysis and they have been referred to each time the results of a bivariate or multivariate analysis are discussed.

The authors conducted a bivariate analysis to select predictors of breastfeeding early initiation and this name should be used throughout instead of “simple” logistic regression. Please mention the level of significance that what was considered in the bivariate analysis to select variables to the multivariate analysis. Also, please clarify why was used a chi-square test and bivariate logistic regression and how both analyses informed the selection of independent variables. Please provide a rationale to use both methods since this is not clear in the methods or results section.

Response: This has been done and stated at the appropriate places of the manuscript.

Please review the English verbal tense throughout. A few parts of the methodology are in present and others present perfect, please choose one and be consistent throughout.

Response: We have red and re-read the manuscript to remove any discrepancy in tense.

Minor comments:

Line 84-91. Please clarify if mothers who did not breastfeed were included or excluded.

Response: The mothers who did not breastfeed were excluded from the analysis. This is stated in the description of the sample, namely “which comprise a weighted sample of 6,616 women who had their most recent births in the preceding two years and who ever breastfed their babies”.

Line 92-109. Why were these predictors chosen? Was this based on literature review or data availability? What is the plausibility of selecting these predictors? Was there any framework used to select variables?

Response: The predictors are chosen based on literature review and data availability. This study is based on secondary analysis of the 2017 Indonesia Demographic and Health Survey, so the choice of the predictors is also based on the data availability. 

Line 110. Remove the word “methods”

Response: Thank you. We have removed it. 

Line 111. Remove the word “basically”

Response: Thank you. We have removed it. 

Line 114 -115. Remove the word “simple” on both lines

Response: Thank you. We have removed it. 

Line 115. Remove the word “simple” logistic regression.

Response: Thank you. We have removed it. 

Results

Overall comment:

The result section is long and repetitive. It has to be streamlined with the description of methodological steps. Tables 1 or 2 should be removed or revised to show a descriptive analysis of the whole population (not per outcome) and bivariate analysis. In the text, authors should focus their report in the most important results not show interpretations, interpretations should be placed in the discussion section. For instance, the authors could highlight important characteristics of the population studied and which were the variables selected in the bivariate to the multivariate analysis.

Line 160 - Remove the word the definition for early breastfeeding initiation. Once this was defined in the method section there is no need for repetition. Also, if authors will opt for using an acronym for early initiation of breastfeeding (EIBF), please define it for the first time in the text and use this throughout.

Response: Thank you. We removed the definition of EIBF since it had already mentioned before. We also have revised the manuscript and used the acronym of EIBF after it is describe clearly in the introduction. 

Tables 1 and 2. Please clarify and justify the need for having these two tables. I strongly recommend removing table 1 and keep with table 2.

Response: Thank you for your suggestion. We have removed Table 1 and kept Tables 2 and 3 of the previous draft, which now become Tables 1 and 2 in the revised draft. 

Line 156. Please remove subheading “Characteristics of respondents”

Response: Thank you. We have removed it. 

Line 220. Please remove subheading “Predictors of Early Initiation Breastfeeding”

Response: Thank you. We have removed it. 

Lines 161-210. Please revise accordingly overall comment above.

Response: Thank you. We have revised it.

Lines 221-228. Consider summarizing it and moving it to the analysis section.

Response: We have revised the paper, and put in the analysis section. 

Lines 235-261. Please summarize and revise accordingly the overall comment above.

Response: Thank you for your suggestion. We have revised the results in the same order with the methods and the discussions.

Table 3 is confusing, please make sure it is necessary to present bivariate analysis results in this table. Again, consider choosing between chi-square and bivariate logistic regression to select variables to multivariate model, detail this step in the methods, and provide a clear cut-off point to include or not a variable in the model.

Response: Thank you for your suggestion. We now use bivariate logistic regression to select the variables as predictors in the multivariate model. Table 2 presents the results of simple and multivariate logistic regression analysis with unadjusted and adjusted odds ratios. We have mentioned clearly in the methods that “The final model includes explanatory variables for which the level of significance is less than or equal to 5 percent (P � 0.05). Odds ratios (OR) are reported with a 95% confidence interval (95% CI).”

Minor comment: please make sure tables follow the same formatting style.

Response: We have tried to make the formatting of the tables as similar as possible to one another.

Discussion:

Overall comment: Consider moving a few sentences from Indonesia context to the introduction, this will help readers to understand the context of the study earlier. Also, was there any hypothesis the authors were trying to explore? Or was this an exploratory study? What gaps this analysis fills?

Response: This study is further analysis of cross-sectional research. This research will answer the research question of “what is the strongest predictors of EIBF”. The result will be useful to strengthen the policy related to EIBF. 

Lines 285-288/298-300/310-312. Do not report results in the discussion.

Response: Thank you. We have revised the paper.

Line 316 – this sentence seems to be incomplete.

Response: Thank you. We have revised the sentences in the revised paper. It is in line 343.

Conclusion: The conclusion lacks to respond to the research question. Why is important to know the predictors of EIBF?

Response: It is important to understand the predictors of EIBF so that appropriate interventions can be designed to strengthen help increase the prevalence breastfeeding for newborn babies. The Conclusion section of the paper has been modified accordingly.

---

## [Decision Letter · Decision Letter 1]

8 Jun 2020

PONE-D-20-03622R1

Predictors of early initiation of breastfeeding in Indonesia: A population-based cross-sectional survey

PLOS ONE

Dear Dr. Gayatri,

Thank you for submitting your manuscript to PLOS ONE. After careful consideration, we feel that it has merit but does not fully meet PLOS ONE’s publication criteria as it currently stands. Therefore, we invite you to submit a revised version of the manuscript that addresses the points raised during the review process.

The authors have made important improvements throughout the manuscript. However, further careful revision is still needed. We are providing a last opportunity for attendance of all reviewer´s requirements. 

We look forward to receiving your revised manuscript.

Kind regards,

Marly A. Cardoso, Ph.D.

Academic Editor

PLOS ONE

Reviewers' comments:

Reviewer's Responses to Questions

**Comments to the Author**

1. If the authors have adequately addressed your comments raised in a previous round of review and you feel that this manuscript is now acceptable for publication, you may indicate that here to bypass the “Comments to the Author” section, enter your conflict of interest statement in the “Confidential to Editor” section, and submit your "Accept" recommendation.

Reviewer #1: (No Response)

Reviewer #2: All comments have been addressed

Reviewer #3: (No Response)

2. Is the manuscript technically sound, and do the data support the conclusions?

Reviewer #1: Yes

Reviewer #2: Yes

Reviewer #3: Partly

3. Has the statistical analysis been performed appropriately and rigorously? 

Reviewer #1: I Don't Know

Reviewer #2: Yes

Reviewer #3: Yes

4. Have the authors made all data underlying the findings in their manuscript fully available?

Reviewer #1: Yes

Reviewer #2: Yes

Reviewer #3: Yes

5. Is the manuscript presented in an intelligible fashion and written in standard English?

Reviewer #1: No

Reviewer #2: Yes

Reviewer #3: Yes

6. Review Comments to the Author

Reviewer #1: The authors responded appropriately to the reviewer's suggestions, especially in the introduction section. However, although the authors have made several changes throughout the text, the methods, results, and discussion section still need to be carefully reviewed. A more appropriate scientific language is also still required. The general comments below may help the authors to improve the manuscript.

-Review the results presented in the abstract: How do those factors impact on the EIBF?

-In the dependent variable paragraph, you used EIBF definitions instead of describing how it was measured, and viceversa. The definition of EIBF is different from how it was collected or measured in the population.

-The description of the independent variables and the results of table 1 and 2 should be presented more succinctly.

-This kind of comment is not appropriate in the result section: “Perhaps children whose fathers are not working have working mothers...”

-Table 1 and 2: Check all the variables and make sure the terms you choose are consistent: for example, “Fathers' occupation” or “Husband's occupation”?

-When you say “As mentioned earlier…” Check if you are repeating information.

-Please, be careful with the acronyms used in tables

-For “Mother working status, skin-to-skin contact, and size at birth” results have not been widely discussed.

-Fathers' occupation (in non-agricultural activities) was not discussed, but it was mentioned as a predictor in this study.

Reviewer #2: The authors attended the suggestions and improved the manuscript substantially. However, in my opinion, some minor points should be still addressed before publication:

Abstract:

- Line 21-22: medical level? The methods section describes only three levels.

Methods:

- In the revised version of the manuscript, the authors used the acronym for delayed initiation of breastfeeding (DIBF) in line 109, but continued using the full name in other sentences (e.g.: line 112). I suggest checking it out.

- Lines 104-105: To make the text more succinct and avoid repetition, I suggest deleting the sentence: “By definition, these children would be aged between 0 and 23 months at the time of the survey.”

Results:

- To make this section more concise and easier to understand, I personally suggest the authors to rewrite the second paragraph (lines 223-251). As it contains only descriptive information, statistical significance should not be attributed (e.g. line 238: “Having skin-to-skin contact soon after birth is associated with a higher prevalence of EIBF.”).

- Lines 236-237 and 263-264: Sentences should be moved to discussion section.

- Table 1: In order to make this table clearer, I suggest deleting the column “Total” (number and %) and line “Total” (1st line) and adding the total number of EIBF and DIBF in the title line [e.g.: Early initiation of breastfeeding (n=3,742) / Delayed initiation of breastfeeding (n=2,874)].

Conclusion:

- I would consider deleting the sentence from lines 418-420 (“This study has been aimed…”) and including a sentence briefly presenting the EIBF predictors found in this study.

Reviewer #3: The revised manuscript brings a clearer description of the analytical approach. However, I have still questions and comments to further improve the manuscript. A detailed comment is below:

Abstract

Lines 25-27 – Please clarify what prominent means. Are these the factors associated with EIBF? If so, report the OR consistently as reported to C-section.

Line 29 - Mothers delivering with C-section were less likely to practise EIBF compared to which group. Please include.

Line 30 - It is not clear in the abstract why skin-to-skin and normal delivery are the strongest factors influencing EIBF. Please report the data in the results to support your conclusions.

Line 31 – replace - early initiation of breastfeeding to EIBF

Results

Results should be summarized to highlight the main findings. In its current form it is very long and not informative. Please be careful to not make inferences in the result section. Eg. Lines 236-237, 234-235, etc. Any inferences if appropriate should be placed in the discussion and corroborated with the literature.

Tables

Tables - Please organize your variables in a consistent order. You should organize your variables within the tables following the description given in the methods – under the categories of individual, individual, household and community level variables.

Table 1 - remove the 100% column. It is unnecessary.

Table 2 – Please double-check the unadjusted OR for working women it seems incorrect.

Conclusions

There were many factors significantly associate with EIBF why authors are issuing recommendations just for skin-to-skin and C-section. Please revise conclusions accordingly specifying the key associated/protective factors that need tackling and whether the counselling strategies should be put in place through hospitals or communities or both?

7. PLOS authors have the option to publish the peer review history of their article (what does this mean?). If published, this will include your full peer review and any attached files.

Reviewer #1: No

Reviewer #2: No

Reviewer #3: No

---

## [Author Response · Author response to Decision Letter 1]

17 Jul 2020

We thank you for your consideration of this resubmission. We appreciate your time and look forward to your response. On the other file, you will find our details of response to the reviewer comments. Thank you.

---

## [Decision Letter · Decision Letter 2]

25 Aug 2020

PONE-D-20-03622R2

Predictors of early initiation of breastfeeding in Indonesia: A population-based cross-sectional survey

PLOS ONE

Dear Dr. Gayatri,

Thank you for submitting your manuscript to PLOS ONE. After careful consideration, we feel that it has merit but does not fully meet PLOS ONE’s publication criteria as it currently stands. Therefore, we invite you to submit a revised version of the manuscript that addresses the points raised during the review process.

The authors have made great efforts for reviewing the manuscript, following most of the reviewers´ recommendations. However, there are still  important points for carefully revision as suggested by the reviewer.

We look forward to receiving your revised manuscript.

Kind regards,

Marly A. Cardoso, Ph.D.

Academic Editor

PLOS ONE

Reviewers' comments:

Reviewer's Responses to Questions

**Comments to the Author**

1. If the authors have adequately addressed your comments raised in a previous round of review and you feel that this manuscript is now acceptable for publication, you may indicate that here to bypass the “Comments to the Author” section, enter your conflict of interest statement in the “Confidential to Editor” section, and submit your "Accept" recommendation.

Reviewer #1: (No Response)

Reviewer #2: All comments have been addressed

2. Is the manuscript technically sound, and do the data support the conclusions?

Reviewer #1: Yes

Reviewer #2: Yes

3. Has the statistical analysis been performed appropriately and rigorously? 

Reviewer #1: I Don't Know

Reviewer #2: Yes

4. Have the authors made all data underlying the findings in their manuscript fully available?

Reviewer #1: Yes

Reviewer #2: Yes

5. Is the manuscript presented in an intelligible fashion and written in standard English?

Reviewer #1: No

Reviewer #2: Yes

6. Review Comments to the Author

Reviewer #1: The authors followed most of the reviewer's suggestions; however, some points still need to be carefully reviewed. The general comments below may guide the authors:

Abstract:

-Is the confidence interval 95%? Please add this information.

-The result and conclusion paragraphs repeat information.

-Please, note that the abstract exceeds the number of words allowed.

Main text:

-The following sentences repeat information: “For more than one child born to a woman in the two years preceding the survey, information pertaining to the most recently born child is analysed in this study.” and “The dependent variable in this study is early initiation of breastfeeding of the woman’s most recent child born in the two years preceding the survey.”

-In the beginning of the dependent variable section you can use the acronym “EIBF” as you have already mentioned it before in the introduction.

-You do not need to abbreviate EIBF in this section again.

-With the following information, please reorganize the dependent variable paragraph (you mixed "definition, measurement and categorization”): 1) Definition of EIBF: “The practice of initiating breastfeeding within the first hour of birth = Early Initiation of Breastfeeding” 2) How was the information on EIBF collected or measured?: “The timing of the initiation of breastfeeding is reported by the women in terms of the number of hours or days after delivery when they started breastfeeding their babies.” 3) Categorization of EIBF for analyses: “The dependent or outcome variable is defined(?) in a binary form, namely EIBF or delayed initiation of breastfeeding (DIBF).” 4) WHO indicator to estimate EIBF: “proportion of infants aged 0-23 months who were put to their mother’s breast within one hour of birth” (WHO. Indicators for assessing infant and young child feeding practices part I: definitions. WHO; 2008.)

-You could consider the next example to present the independent variables: “Data were obtained from the hospital records, which included gestational age at delivery (<37 or >=37 weeks), type of delivery (vaginal or caesarean), child’s sex (female or male)”. This way you will have a more succinct paragraph.

-You use child’s size at birth as a proxy for birth weight, but what information was used to classify babies as large or small (small/large for gestational age)? I could not understand this variable.

-Thanks for the description of the wealth index, but please summarize the paragraph. It is important to mention briefly where it comes from and its reference.

-The ‘svy’ command explanation is not necessary.

-The main characteristics of the study population should be presented succinctly (lines 226 to 261).

-This kind of comment is not appropriate in the result section: “Women who give birth to small sized babies are probably themselves undernourished compared to the women who give birth to average or large sized babies.”

-What is the reference for the following information? “…although it is higher than that in Vietnam (27%), Thailand 312 (40%) and Lao People’s Democratic Republic (50%)”

-The information presented from line 328 to 343 I do not think is necessary.

-“In terms of education however, studies in other countries have shown...” which countries?

-“This apparent anomaly may be explained by…” what do you mean by anomaly?

- Please, avoid “(table 1)” in discussion section.

-What does “[12 Figure 3.3]” mean? Line 357.

-Is this information related to EIBF? “working women might be concerned about their professional and body image, so they might consider breastfeeding an obstacle to return to their physical shape of the time before they became pregnant [18,24].”

-This information is repeated in the text:

“the IDHS 2017 data were collected by using a two-stage stratified sampling.”

“As mentioned earlier, the data used in this research were collected at the 2017 Indonesia Demographic and Health Survey.”

“…had at least one live birth during the two years preceding the survey and who breastfed their most recently born child.”

“confidence interval (CI)”

“the Indonesia Demographic and Health Survey (IDHS)”

“(proxy for birth weight)”.

“the occupation of women’s husbands (father’s occupation) is found to be associated with EIBF in a manner that is different from the association of EIBF with women’s occupation.” (similar words)

“as mentioned earlier, EIBF has benefits for the health of both mothers and infants”

“early initiation of breastfeeding (EIBF)”

“(see Table 1)” = “The categories of each variable are given in Table 1”.

Reviewer #2: The authors made several improvements in the text. The points I raised in response to the last submission of this manuscript have been sufficiently addressed. I consider that the article is now suitable for publication in this Journal.

7. PLOS authors have the option to publish the peer review history of their article (what does this mean?). If published, this will include your full peer review and any attached files.

Reviewer #1: No

Reviewer #2: No

---

## [Author Response · Author response to Decision Letter 2]

5 Sep 2020

Review Comments to the Author

Reviewer #1: The authors followed most of the reviewer's suggestions; however, some points still need to be carefully reviewed. The general comments below may guide the authors:

Abstract:

-Is the confidence interval 95%? Please add this information.

Response: Thank you. Yes, it is 95% confidence interval. We have specified “95% CI” in the abstract section, as advised. 

-The result and conclusion paragraphs repeat information.

Response: Thank you. We have revised the conclusion. 

-Please, note that the abstract exceeds the number of words allowed.

Response: Thank you. We have revised the abstract, which has less than 300 words now.

Main text:

-The following sentences repeat information: “For more than one child born to a woman in the two years preceding the survey, information pertaining to the most recently born child is analysed in this study.” and “The dependent variable in this study is early initiation of breastfeeding of the woman’s most recent child born in the two years preceding the survey.”

Response: Thank you. We have removed the first sentence and kept the second sentence.

-In the beginning of the dependent variable section you can use the acronym “EIBF” as you have already mentioned it before in the introduction.

-You do not need to abbreviate EIBF in this section again.

Response: Thank you. We have kept the first mention of “Early Initiation of Breastfeeding (EIBF)” in the Introduction and used only the acronym thereafter. 

-With the following information, please reorganize the dependent variable paragraph (you mixed "definition, measurement and categorization”): 1) Definition of EIBF: “The practice of initiating breastfeeding within the first hour of birth = Early Initiation of Breastfeeding” 2) How was the information on EIBF collected or measured?: “The timing of the initiation of breastfeeding is reported by the women in terms of the number of hours or days after delivery when they started breastfeeding their babies.” 3) Categorization of EIBF for analyses: “The dependent or outcome variable is defined(?) in a binary form, namely EIBF or delayed initiation of breastfeeding (DIBF).” 4) WHO indicator to estimate EIBF: “proportion of infants aged 0-23 months who were put to their mother’s breast within one hour of birth” (WHO. Indicators for assessing infant and young child feeding practices part I: definitions. WHO; 2008.)

Response: Thank you. We have reorganized the dependent variable. 

-You could consider the next example to present the independent variables: “Data were obtained from the hospital records, which included gestational age at delivery (<37 or >=37 weeks), type of delivery (vaginal or caesarean), child’s sex (female or male)”. This way you will have a more succinct paragraph.

Response: Thank you for your suggestions. We have revised the independent.

-You use child’s size at birth as a proxy for birth weight, but what information was used to classify babies as large or small (small/large for gestational age)? I could not understand this variable.

Response: Each woman who had ever given birth was asked the following question with respect to her two most recent births: “When (NAME) was born, was (NAME) very large, larger than average, average, smaller than average, or very small”? (Question 426, Woman’s Questionnaire, Appendix A, Indonesia Demographic and Health Survey 2017, Final Report p.475). This was done especially when the birth took place at home where there was no facility to weigh the child at birth (National Population and Family Planning Board, Statistics Indonesia, Ministry of Health, & ICF, 2018, p.160). The child’s size at birth was thus recorded according to the woman’s perception of her child’s size at birth. The woman’s response of the child being “very small” or “smaller than average,” was taken to be a proxy for Low Birth Weight. 

-Thanks for the description of the wealth index, but please summarize the paragraph. It is important to mention briefly where it comes from and its reference.

Response: Thank you. We have mentioned the source from the DHS Program.

-The ‘svy’ command explanation is not necessary.

Response: Thank you. We have removed it in the method section. 

-The main characteristics of the study population should be presented succinctly (lines 226 to 261).

Response: Thank you, we have revised it. 

-This kind of comment is not appropriate in the result section: “Women who give birth to small sized babies are probably themselves undernourished compared to the women who give birth to average or large sized babies.”

Response: Thank you. We have removed it. 

-What is the reference for the following information? “…although it is higher than that in Vietnam (27%), Thailand 312 (40%) and Lao People’s Democratic Republic (50%)”

Response: The data from the UNICEF. We have added the reference.

-The information presented from line 328 to 343 I do not think is necessary.

Response: Thank you. However, we think that this part is important for providing clarifications about those predictors that have no association with EIBF in Indonesia. 

-“In terms of education however, studies in other countries have shown...” which countries?

Response: in Ethiopia and Malawi (Liben & Yesuf, 2016; Nkoka, Ntenda, Kanje, Milanzi, & Arora, 2019).

-“This apparent anomaly may be explained by…” what do you mean by anomaly?

Response: By “anomaly” we mean “difference or “irregularity”. We have re-written the sentence and replaced the word “anomaly’ with the word “difference”

- Please, avoid “(table 1)” in discussion section.

Response: Thank you. We have removed “table 1” in discussion section. 

-What does “[12 Figure 3.3]” mean? Line 357.

Response: We have removed it. Thank you. 

-Is this information related to EIBF? “working women might be concerned about their professional and body image, so they might consider breastfeeding an obstacle to return to their physical shape of the time before they became pregnant [18,24].”

Response: It might be related. Working women who are concerned about their body image might choose not to breastfeed their babies, or choose to curtail their breastfeeding period, thus adversely affecting EIBF and exclusive breastfeeding practices. We have provided references to the sources of such a statement.

-This information is repeated in the text:

“the IDHS 2017 data were collected by using a two-stage stratified sampling.”

Response: Thank you, we have removed the repetition. 

“As mentioned earlier, the data used in this research were collected at the 2017 Indonesia Demographic and Health Survey.”

Response: Thank you, we have removed the repetition. 

“…had at least one live birth during the two years preceding the survey and who breastfed their most recently born child.”

Response: Thank you, we have removed the repetition. 

“confidence interval (CI)”

Response: Thank you, we have removed the repetition. 

“the Indonesia Demographic and Health Survey (IDHS)”

Response: Thank you, we have kept it where it is mentioned for the first time in the main text, but removed the repetition thereafter. 

“(proxy for birth weight)”

Response: Thank you, we have removed the repetition. 

“the occupation of women’s husbands (father’s occupation) is found to be associated with EIBF in a manner that is different from the association of EIBF with women’s occupation.” (similar words)

Response: Thank you, we have removed it due to repetition. It has been explained in the next sentence.

“as mentioned earlier, EIBF has benefits for the health of both mothers and infants”

Response: Thank you, we have removed the repetition. 

“early initiation of breastfeeding (EIBF)”

Response: Thank you, we have removed the repetition. 

“(see Table 1)” = “The categories of each variable are given in Table 1”.

Response: Thank you, we have revised the sentence as you have suggested. 

Reviewer #2: The authors made several improvements in the text. The points I raised in response to the last submission of this manuscript have been sufficiently addressed. I consider that the article is now suitable for publication in this Journal.

Response: Thank you.

7. PLOS authors have the option to publish the peer review history of their article (what does this mean?). If published, this will include your full peer review and any attached files.

Do you want your identity to be public for this peer review? For information about this choice, including consent withdrawal, please see our Privacy Policy.

Reviewer #1: No

Reviewer #2: No

---

## [Editor Report · Decision Letter 3]

8 Sep 2020

Predictors of early initiation of breastfeeding in Indonesia: A population-based cross-sectional survey

PONE-D-20-03622R3

Dear Dr. Gayatri,

We’re pleased to inform you that your manuscript has been judged scientifically suitable for publication and will be formally accepted for publication once it meets all outstanding technical requirements.

Kind regards,

Marly A. Cardoso, Ph.D.

Academic Editor

PLOS ONE
---

## [Editor Report · Acceptance letter]

16 Sep 2020

PONE-D-20-03622R3 

Predictors of early initiation of breastfeeding in Indonesia:A population-based cross-sectional survey 

Dear Dr. Gayatri:

I'm pleased to inform you that your manuscript has been deemed suitable for publication in PLOS ONE. Congratulations! Your manuscript is now with our production department. 

Kind regards, 

on behalf of

Dr. Marly A. Cardoso 

Academic Editor

PLOS ONE